# PHYSICS-GROUNDED MOTION FORECASTING VIA EQUATION DISCOVERY FOR TRAJECTORY-GUIDED IMAGE-TO-VIDEO GENERATION

## ABSTRACT

Recent advances in video generation models have achieved remarkable visual realism. However, these models typically lack accurate physical alignment, failing to replicate real-world dynamics in object motion. This limitation arises primarily from their reliance on learned statistical correlations rather than capturing mechanisms adhering to physical laws. To address this issue, we introduce a novel framework that integrates symbolic regression (SR) and trajectory-guided image-to-video (I2V) models for physics-grounded video forecasting. Our approach extracts motion trajectories from input videos, uses a retrieval-based pre-training mechanism to enhance symbolic regression, and discovers equations of motion to forecast physically accurate future trajectories. These trajectories then guide video generation without requiring fine-tuning of existing models. We evaluate our framework on scenarios from classical mechanics, including spring-mass, pendulums, and projectile motions. In these settings, our method successfully recovers ground-truth analytical equations and improves the physical alignment of generated videos compared to baseline methods. This work provides a first step toward integrating equation discovery with video generation.[1]

## 1 INTRODUCTION

Recent advances in video generation models have significantly improved the realism of synthesized videos, driven primarily by diffusion-based and autoregressive models Blattmann et al. (2023); Yang et al. (2025); Team (2025); Kong et al. (2025). Incorporating motion trajectories enables precise control over object movements, facilitating videos that more accurately capture intended dynamics Wu et al. (2025); Wang et al. (2024b); Namekata et al. (2025). However, existing trajectory-guided methods typically rely on text prompts, manually drawn or statistically derived trajectories Zhang et al. (2024); Team, none of which ensures adherence to the underlying laws of physics Kang et al. (2024); Motamed et al. (2025); Wang et al. (2025).

Physicists understand object dynamics by discovering physical laws from observational data and formulating these laws into symbolic equations. These equations reliably forecast object movements, unaffected by shifts in the underlying data distributions. Moreover, such equation discovery does not require extensive training data, unlike the scaling laws commonly adopted by current video generation models Kaplan et al. (2020). Therefore, for the *first* time, we investigate: i) whether AI methods can feasibly discover physics equations directly from video clips and subsequently use these equations to reliably forecast object motion trajectories, and ii) whether such equations can be identified from just one or a handful of video clip without extensive data-driven training.

To address the above research questions, we propose a novel *neuro-symbolic, inference-only* framework for forecasting object motion trajectories from a short video clip, followed by feeding the predicted trajectories into an image-to-video (I2V) model to produce physics-grounded videos. As illustrated in Figure 1, our approach first utilizes CoTracker Karaev et al. (2024) to extract initial object motion trajectories from a short video clip. We then employ a symbolic regression (SR) algorithm Cranmer (2023a), an evolutionary search method that automatically discovers explicit

---

[1]The code and dataset are available at `https://anonymous.4open.science/r/ReSR-0083/`

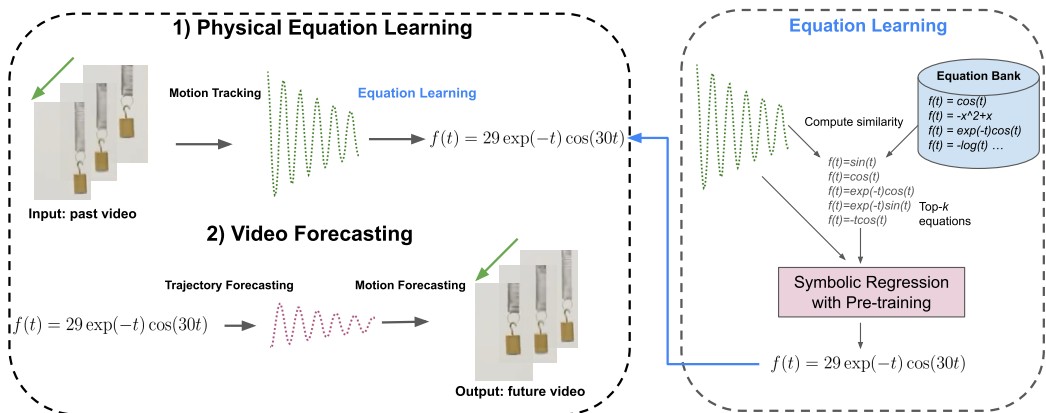

Figure 1: An overview of our framework. Given an input video, we first extract object (*i.e.,* spring or weight) motion trajectories, which are used to discover equations of motion via SR enhanced by our proposed retrieval-based pre-training mechanism (ReSR), where ReSR initializes the search with candidates retrieved from a curated equation bank of known physical laws. The learned symbolic equations forecast future object trajectories, serving as precise control signals to guide trajectory-guided video generation models, resulting in more physics-grounded video generation.

mathematical equations, to derive a *human-interpretable* symbolic equation characterizing the underlying physical law. Given the initial trajectories, this discovered equation can reliably produce future object movements of arbitrary length, consistently adhering to the underlying physics laws.

From another perspective, the equation discovery process can be viewed as training a symbolic model that characterizes motion trajectories. Current evolutionary search methods typically initialize their searches using randomly selected functions, often starting far from the global optimum and resulting in slow convergence. To mitigate this, we propose a novel **Re**trieval-based pre-training method for **S**ymbolic **R**egression, called ReSR, which initializes the search from relevant equations retrieved from a physical equation bank. Unlike prior SR methods that rely on randomly initialized function sets, ReSR incorporates physics-inspired equations, reducing search space bias and improving both efficiency and interpretability by aligning candidates with established physics priors.

To investigate the fundamental challenges of learning equations from given videos, we conduct experiments on a set of videos in a controlled laboratory environment governed by the laws of classical mechanics. These videos depict systems, such as spring-mass oscillators, pendulums, and projectile motion. We choose this controlled setting because: i) it enables direct evaluation of discovered equations against ground-truth equations identified by physicists; ii) insights into object motion in classical mechanics can be easily extended into other types of motion; and iii) classical mechanics underpins a wide range of real-world applications, including physics simulation, scientific visualization, and physics education.

Our contributions are summarized as follows:

- We propose a novel neuro-symbolic framework for physics-grounded video forecasting. Specifically, our method first extracts motion trajectories from input videos, then discovers equations of motion. These equations are used to forecast future trajectories, which then guide I2V models to synthesize future videos that better align with physical laws. Importantly, our approach operates entirely at inference time and does not require fine-tuning of video generation models.

- We introduce a *retrieval-based pre-training mechanism* for SR, denoted as ReSR, which leverages a curated equation bank of known physical laws to provide strong initialization candidates. This substantially improves convergence speed and accuracy in discovering equations from observed trajectories.

- We empirically demonstrate that our framework not only recovers equations closely aligned with ground-truth analytical expressions and observed trajectories, but also generates videos with significantly improved physical consistency compared to existing baselines when conditioned on trajectories discovered by ReSR.

## 2 PRELIMINARY

Scientists have discovered empirical laws from observational data. For example, Johannes Kepler formulated the third law of planetary motion, $(\text{period})^2 \propto (\text{radius})^3$, after analyzing thirty years of astronomical data. Similarly, Planck's law was a function fitted to experimental data Planck (1900). However, modern scientific data is often high-dimensional and complex, making manual equation discovery a challenging task Virgolin & Pissis (2022). SR is a computational method for automatically deriving mathematical equations from data. Unlike traditional regression, which fits data to a predefined equation structure (*e.g.,* linear or polynomial regression), SR searches for both the equation structure and parameters. This flexibility makes SR particularly valuable in scientific discovery Rudy et al. (2017); Meidani & Farimani (2023).

SR approaches can be broadly categorized into two primary types: evolutionary algorithm (EA)-based methods and deep learning-based methods. EA-based approaches operate by evolving a population of candidate equations over successive generations, using operations such as mutation and crossover to search for equations that best fit the data Brindle (1980); Goldberg & Deb (1991); Zhang & Shasha (1989); Stephens (2025); Cranmer (2023b). EA-based methods require minimal prior assumptions about equation structure, allowing them to explore a diverse space.

Deep learning-based methods directly predict equations from data Biggio et al. (2021); Kamienny et al. (2022); Shojaee et al. (2023); Meidani et al. (2024). Devlin et al. (2019); Radford et al. (2019); Feng et al. (2023; 2025) typically train an end-to-end transformer-based model where the input is observational data and the output is a symbolic equation. However, deep learning models often struggle with out-of-distribution generalization Yang et al. (2024); Kim et al. (2024); Feng et al. (2024), and cannot guarantee the generated output forms a syntactically valid equation, leading to non-executable equations. Inspired by the success of pre-training in deep learning Erhan et al. (2010); Devlin et al. (2019), we propose a pre-training mechanism for EA-based SR (see Section 3.3). We first construct an equation bank containing physics-related equations. During initialization, the SR algorithm retrieves equations that closely align with the observed data and uses them as initial candidates. This pre-training strategy significantly improves convergence speed and enhances the accuracy of the learned equations.

## 3 METHODOLOGY

### 3.1 TASK FORMULATION AND NOTATIONS

The objective of this study is to achieve physics-grounded motion forecasting for trajectory-guided video generation. As illustrated in Figure 1, given an input video $V_i$ depicting the initial motion of an object, our approach generates a video $V_o$ representing the object's future motion. Our approach consists of three main steps. First, we extract the motion trajectories of moving objects in $V_i$. The extracted trajectories are represented as a set $\mathbb{P} = \{p_1, p_2, ..., p_n\}$, where each trajectory $p_i$ is a time series of object positions: $p_i = [p_1, p_2, ..., p_T]$, where $p_t = (x_t, y_t)$ denotes the image-space coordinate of the object at time step $t$. Next, we employ symbolic regression to learn equations that govern the motion of objects. Specifically, for each trajectory $p_i$, we aim to learn a pair of functions $f_i^x(t)$ and $f_i^y(t)$ such that:

$$x_t = f_i^x(t), \quad y_t = f_i^y(t). \tag{1}$$

that map time to object position. Using the learned equation $f_i^x(t)$ and $f_i^y(t)$, we predict the future trajectory for time steps beyond the observed interval, i.e., $p_i = f_i(t), t \in \{T+1, T+2, ..., T+K\}$, where $K$ represents the forecast horizon. Finally, we utilize the predicted trajectories to guide trajectory-based video generation models, which then synthesize the future video $V_o$.

### 3.2 EXTRACTION OF OBJECT MOTION TRAJECTORY

To learn equations of object motion, we first extract object trajectories from the input video $V_i$. We employ CoTracker Karaev et al. (2024), a state-of-the-art point tracking model that performs joint point tracking and propagation across all frames. CoTracker requires a set of query points in the first frame to initiate tracking. While manual annotation is possible, it is not scalable across diverse video content. Instead, we adopt an automated approach by uniformly sampling query points on a 2D $M \times M$ grid across the first frame. Each query point is tracked throughout the entire video.

We perform all tracking in the original image coordinate system without additional preprocessing. After collecting all trajectories, we compute the temporal variance of each trajectory. We then rank the trajectories based on their positional variance across time and retain the top-$k$ trajectories with the highest motion magnitude. This strategy is motivated by the observation that target objects in physics-driven videos typically exhibit the most motion, while background regions tend to remain static. As a result, selecting high-variance trajectories increases the likelihood of capturing the true object dynamics and filtering out irrelevant background noise.

### 3.3 SYMBOLIC REGRESSION WITH PRE-TRAINING

In this step, we apply symbolic regression with retrieval-based pre-training (ReSR) to discover equations that fit extracted object trajectories. Instead of initializing the search process from scratch with random equations, we retrieve a set of candidate equations from a curated equation bank composed of physics-related equations. The retrieved equations then serve as priors to initialize the symbolic regression. Given a trajectory $\boldsymbol{p}_i = [p_1, p_2, ..., p_T]$, our goal is to learn Equation 1.

**Construction of Equation Bank.** We construct an equation bank containing a diverse set of equations derived from classical and empirical physics to serve as priors for symbolic regression. The bank integrates equations from three sources: 1) The Feynman equation dataset Udrescu & Tegmark (2020), which consists of equations extracted from the Feynman Lectures on Physics Feynman et al. (1965). These equations typically take the form $y = f(x_1, x_2, \dots)$, with up to ten input variables. To adapt them for time-series motion, we substitute time-dependent variables (*e.g.,* velocity, acceleration, momentum) with the time variable $t$. Variables that are independent of time (*e.g.,* mass, density) are replaced with constant values (*e.g.,* 10), aiming to preserve equation structure. We select a total of 106 equations after this adaptation. 2) The Nguyen dataset Uy et al. (2011), which includes 10 commonly used empirical formulas in symbolic regression benchmarks. We apply the same time-variable substitution process. 3) We include 13 additional physics equations from Thornton & Marion (2004), not present in the aforementioned datasets, to ensure the equation bank represents a broad range of physical systems. All equations are stored as symbolic expressions in Julia syntax Bezanson et al. (2017), enabling compatibility with our symbolic regression framework.

**Retrieval-based Pre-training Mechanism.**

Our proposed ReSR initializes symbolic regression with candidate equations retrieved from a curated equation bank. The retrieval is based on the similarity between the extracted object trajectory and trajectories generated by each equation in the bank. Similarity is computed using Dynamic Time Warping (DTW) Müller (2007), a sequence alignment algorithm that handles temporal misalignments such as phase shifts and local time warping that are not captured by Euclidean distance.

However, standard DTW is unable to robustly handle spatial offsets and scale variations in trajectory coordinates. To address this, we introduce *Normalized Dynamic Time Warping* (N-DTW), where the extracted trajectory is rescaled to match the coordinate range of each equation-generated trajectory before computing DTW. This helps the comparison to focus on shape similarity rather than absolute position. Formally, given an extracted trajectory $\boldsymbol{p}_i = [p_1, p_2, ..., p_T]$, where each $p_t = (x_t, y_t)$, we normalize it as follows:

$$\bar{x}_t = (\hat{x}_{\max} - \hat{x}_{\min}) \cdot \frac{x_t - x_{\min}}{x_{\max} - x_{\min}} + \hat{x}_{\min} \tag{2}$$

$$\bar{y}_t = (\hat{y}_{\max} - \hat{y}_{\min}) \cdot \frac{y_t - y_{\min}}{y_{\max} - y_{\min}} + \hat{y}_{\min} \tag{3}$$

where $x_{\min}, x_{\max}, y_{\min}, y_{\max}$ are the bounds of the extracted trajectory, and $\hat{x}_{\min}, \hat{x}_{\max}, \hat{y}_{\min}, \hat{y}_{\max}$ are the bounds of the equation-generated trajectory.

For each equation in the bank, we compute an N-DTW score with the normalized extracted trajectory. Since the similarity between the extracted trajectory and each equation-generated trajectory is computed independently, N-DTW retrieval can be easily parallelized across multiple CPU cores, enabling scalability to large equation banks. We then select the top-$k$ equations with the lowest distances as initial candidates for symbolic regression. This retrieval strategy emphasizes shape similarity rather than proximity in raw values. For instance, consider a trajectory generated by $y = 0.5 \cos(t + 3) + 100$. Two candidate equations might be $y = 100$ and $y = \cos(t)$. While Euclidean distance may favor $y = 100$ due to its proximity in magnitude, it fails to capture the

oscillatory structure. In contrast, N-DTW correctly identifies $y = \cos(t)$ as the more structurally similar trajectory.

**Initialization of ReSR.** We initialize a portion of population members with the top-$k$ retrieved equations that closely match the target trajectory. We introduce an initialization weight hyperparameter $\alpha \in [0, 1]$, which determines the proportion of initial population members that are seeded with retrieved equations, while the remaining are randomly generated. This hybrid initialization strategy allows us to balance *exploration*—via randomly sampled equations that enable diversity in the search space—and *exploitation*—via retrieved equations that act as informative priors. Higher values of $\alpha$ prioritize faster convergence, while lower values preserve the capacity for discovering novel equation forms. If the available number of top-$k$ retrieved equations is insufficient to meet the required number based on $\alpha$, we duplicate top-$k$ retrieved equations to fill the remaining positions. This strategy ensures that the initial population predominantly contains equations closely matching the observed dynamics, reducing the risk of including irrelevant or misleading equations that could negatively impact the search efficiency. This initialization occurs only once at the beginning of the symbolic regression run. All modifications, including retrieval-based pre-training and the integration of N-DTW, are implemented within a modified version of the `SymbolicRegression.jl` framework Cranmer (2023b), ensuring compatibility with existing symbolic regression workflows and reproducibility of our method.

### 3.4 Trajectory-Guided Video Forecasting

To generate future video frames $V_o$ that are physically consistent with learned motion dynamics, we incorporate existing trajectory-guided I2V models, such as SG-I2V Namekata et al. (2025), Tora Zhang et al. (2024), and MotionCtrl Wang et al. (2024b), into our framework. These models are typically diffusion models Song et al. (2020) that synthesize temporally coherent video sequences by denoising noise-perturbed images conditioned on a starting image and motion trajectories.

We use the final frame of the observed input video $V_i$ as the starting image and condition on future trajectories predicted by equations learned from ReSR. These trajectories are formatted as sequences of $(x, y)$ coordinates, sampled at temporal intervals that match the requirement of each I2V model. This integration enables our framework to produce future video sequences that are not only visually plausible but also governed by equations of motion inferred from past observations. Our approach is *modular* and *model-agnostic*: it can be directly applied to any trajectory-guided I2V model without retraining or fine-tuning.

## 4 Experiments

We first assess whether the proposed ReSR enhances the performance of symbolic regression in discovering equations. Then, we examine whether trajectories predicted by the learned equations lead to videos that better align with real-world physical dynamics.

### 4.1 Evaluation of Equation Discovery

**Datasets.** We evaluate equation discovery methods using trajectories extracted from videos of classical physics systems, divided into two categories: **1)** *systems with ground-truth trajectory equations* (*i.e.,* systems with analytical solutions), including spring mass, damped spring mass, two body, and projectile motion Huang et al. (2024); **2)** *systems without ground-truth trajectory equations*, including single pendulum, double pendulum and fluid motion, where no closed-form analytical solution exists Huang et al. (2024); Ohana et al. (2024). Each system includes ten videos with varying initial states. We use CoTracker Karaev et al. (2024) to extract uniformly sampling query points on a $10 \times 10$ grid from the first frame. From these, we select the top 5 trajectories with the highest temporal variance to serve as inputs for symbolic regression methods. Each trajectory is split 80%/10%/10% along the time dimension for training, validation, and evaluation, respectively. This split aims to select equations that generalize from past states to unseen future states.

**Evaluation.** For systems with ground-truth equations, we evaluate symbolic similarity between predicted equations and ground-truth equations using normalized Tree Edit Distance (TED) Zhang & Shasha (1989), which measures how many edit operations (*i.e.,* insertions, deletions, substitutions)

| Methods | **Baseline Comparison** | | | |
| | with AS | | w/o AS | Conv. |
| | TED ($\uparrow$) | MSE ($\downarrow$) | MSE ($\downarrow$) | ITB($\downarrow$) |
|---|---|---|---|---|
| APO | $0.33_{0.11}$ | $7.92_{0.21}$ | $76.52_{7.56}$ | $68.21_{16.51}$ |
| gplearn | $0.40_{0.11}$ | $3.87_{0.21}$ | $61.95_{7.44}$ | $83.14_{12.97}$ |
| uDSR | $0.41_{0.12}$ | $3.73_{0.11}$ | $50.35_{6.10}$ | $74.83_{13.30}$ |
| KAN | $0.22_{0.14}$ | $11.14_{0.49}$ | $91.43_{9.57}$ | N/A |
| PySR | $0.47_{0.16}$ | $2.95_{0.05}$ | $45.25_{4.39}$ | $61.43_{10.37}$ |
| LaSR | $0.54_{0.15}$ | $1.91_{0.05}$ | $32.56_{4.05}$ | $59.93_{11.93}$ |
| **ReSR (Our)** | $\mathbf{0.80_{0.08}}$ | $\mathbf{1.52_{0.04}}$ | $\mathbf{27.58_{3.61}}$ | $\mathbf{44.31_{7.61}}$ |
| **Ablation Study** | | | | |
| Varying Initialization Weight $\alpha$ | | | | |
| ReSR-0 | $0.47_{0.16}$ | $2.95_{0.05}$ | $45.25_{4.39}$ | $61.43_{10.37}$ |
| ReSR-0.25 | $0.60_{0.13}$ | $1.84_{0.06}$ | $30.90_{3.16}$ | $57.24_{8.29}$ |
| ReSR-0.5 | $0.70_{0.12}$ | $1.74_{0.05}$ | $28.19_{3.12}$ | $49.58_{9.82}$ |
| ReSR-0.75 | $0.80_{0.08}$ | $1.52_{0.04}$ | $27.58_{3.61}$ | $44.31_{7.61}$ |
| ReSR-1.0 | $0.77_{0.10}$ | $1.55_{0.04}$ | $28.32_{2.90}$ | $46.63_{8.94}$ |
| **Ablation Study** | | | | |
| Varying Training/Test Split Proportion ($\alpha = 0.75$) | | | | |
| ReSR-2:7 | $0.54_{0.17}$ | $31.96_{5.93}$ | $137.04_{9.17}$ | $33.76_{5.14}$ |
| ReSR-4:5 | $0.61_{0.14}$ | $18.41_{2.32}$ | $89.28_{5.20}$ | $37.66_{5.07}$ |
| ReSR-6:3 | $0.68_{0.13}$ | $7.19_{0.21}$ | $57.51_{6.47}$ | $42.83_{6.39}$ |
| ReSR-8:1 | $0.80_{0.08}$ | $1.52_{0.04}$ | $27.58_{3.61}$ | $44.31_{7.61}$ |

Table 1: Quantitative comparison with baselines and ablation study of ReSR. AS indicates analytical solutions; Conv. indicates convergence. Best results in **bold**.

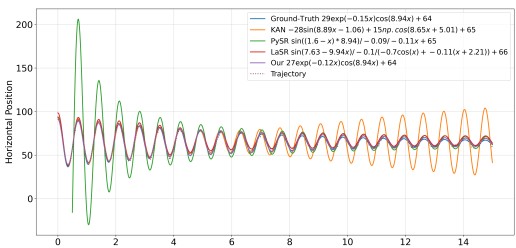

(a) Damped spring–mass system.

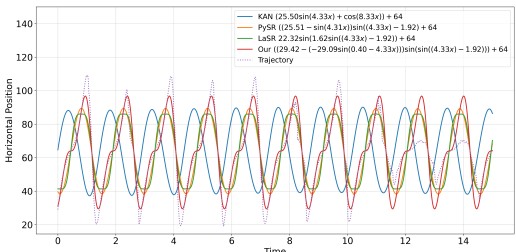

(b) Single pendulum system.

Figure 2: Qualitative comparison of equations discovered by different methods.

are required to transform one equation tree into another, normalized by the maximum node count of two equation trees. For systems without ground-truth equations, we measure the Mean Squared Error (MSE) between the trajectory generated by predicted equations and the actual observed trajectory. To compare the convergence speed of different methods, we report the *iteration-to-best* (ITB) metric, which measures the number of iterations required to reach the method's lowest MSE on the validation set Xing et al. (2018); Smith & Topin (2019).

**Baselines.** We compare against the following methods: **APO** Schmidt & Lipson (2010): A symbolic regression method using Age-fitness Pareto Optimization. **gplearn** Stephens (2025): An EA-based symbolic regression with a scikit-learn-style API. **uDSR** Landajuela et al. (2022): A hybrid approach that combines deep learning models with evolutionary algorithms to discover equations. **KAN** Liu et al. (2025): Kolmogorov-Arnold Networks (KANs) replace each weight in Multi-Layer Perceptrons (MLPs) with a univariate function parameterized as a spline, enabling symbolic equation extraction after training. **PySR** Cranmer (2023b): A symbolic regression framework based on evolutionary search, which can be viewed as an ablation model without retrieval-based pre-training. **LaSR** Grayeli et al. (2024): A symbolic regression approach that leverages large language models to propose initial equations.

**Implementation Details.** For EA-based symbolic regression methods, including both our method and baselines, we run 100 iterations with a population size of 30 across 30 populations. The search space operators include basic arithmetic (+, -, *, /), power functions, and common mathematical functions: cos, sin, exp, log, tan, and sqrt. For KAN, we perform grid-based hyperparameter tuning and report results using the best-performing configuration on the validation set. All experiments are conducted on a machine with 32-core CPUs and a single 80GB A100 GPU.

**Results and Analysis.** Table 1 presents the comparison between ReSR, baseline methods, and ablation variants. ReSR consistently outperforms all baselines in both symbolic similarity (TED) and trajectory error (MSE), demonstrating improved accuracy in discovering physical equations. Additionally, it achieves the fastest convergence (lowest ITB), highlighting the effectiveness of retrieval-based pre-training. For the ablation study, we first analyze the effect of the initialization weight hyperparameter $\alpha$. Performance improves steadily as $\alpha$ increases, peaking at $\alpha = 0.75$, which supports both exploitation (using physics-aligned priors) and exploration (diversity through random sampling). Another ablation study investigates the impact of varying the training/test split while keeping the validation set fixed at 10% of the data. Results show that increasing the training set size improves equation discovery accuracy, indicating that ReSR benefits from larger datasets to better fit observational data.

Figure 2 presents case studies on damped spring-mass and single pendulum systems. ReSR generates trajectories that closely align with the observed data, while baseline methods often produce distorted or phase-shifted results. In the damped spring-mass system, ReSR successfully recovers an equation that matches the ground truth.

## 4.2 EVALUATION OF VIDEO FORECASTING

**Datasets.** We evaluate motion forecasting for video generation using the same set of physical systems described in Section 4.1. Each video generation model takes an initial image (serving as the first frame) and, optionally, a predicted trajectory. Initial images are sourced from both synthetic and real-world domains. Synthetic images are rendered using physics simulators Huang et al. (2024); Ohana et al. (2024) and include systems such as spring mass, damped spring mass, two-body, projectile, and fluid motion. Real initial images are extracted from videos of real-world single and double pendulum systems. For each system, we extract ten initial images, each corresponding to the first frame of the final 10% (test segment) of its video. We apply the learned equations on the training portion (first 80%) of the trajectory and forecast motion into the final 10% segment.

| Models | Visual | | Physical | | Visual | | Physical | |
|---|---|---|---|---|---|---|---|---|
| | FVD($\downarrow$) | FID($\downarrow$) | Smo.($\uparrow$) | TraEr($\downarrow$) | FVD($\downarrow$) | FID($\downarrow$) | Smo.($\uparrow$) | TraEr($\downarrow$) |
| | **Real** | | | | **Synthetic** | | | |
| SVD | $2521_{213}$ | $396_{47}$ | $91.52_{2.31}$ | $632_{134}$ | $1947_{294}$ | $415_{45}$ | $95.24_{2.41}$ | $624_{121}$ |
| CogvidX | $2203_{215}$ | $356_{24}$ | $92.40_{2.34}$ | $512_{84}$ | $1547_{154}$ | $342_{31}$ | $97.38_{1.49}$ | $589_{93}$ |
| Cosmos | $2453_{295}$ | $381_{56}$ | $91.46_{4.35}$ | $552_{78}$ | $1870_{215}$ | $388_{41}$ | $96.39_{1.40}$ | $658_{74}$ |
| Hunyuan | $2382_{274}$ | $387_{52}$ | $92.35_{3.41}$ | $578_{87}$ | $1753_{186}$ | $363_{39}$ | $96.38_{1.57}$ | $562_{63}$ |
| ID | $2607_{221}$ | $423_{31}$ | $86.38_{4.31}$ | $563_{46}$ | $1994_{127}$ | $348_{43}$ | $89.39_{5.95}$ | $567_{81}$ |
| Our$_{DA}$ | $1729_{133}$ | $277_{35}$ | $92.38_{2.10}$ | $489_{39}$ | $853_{142}$ | $248_{35}$ | $97.84_{1.01}$ | $374_{32}$ |
| Our$_{MCtrl}$ | $1793_{148}$ | $266_{31}$ | $91.44_{3.08}$ | $484_{31}$ | $847_{153}$ | $244_{27}$ | $97.43_{1.45}$ | $378_{45}$ |
| Our$_{SG}$ | $1778_{145}$ | $254_{31}$ | $92.39_{3.84}$ | $455_{40}$ | $791_{138}$ | $235_{24}$ | $96.53_{1.46}$ | $384_{33}$ |
| Our$_{Tora}$ | $1674_{133}$ | $235_{34}$ | $95.30_{2.28}$ | $431_{37}$ | $728_{121}$ | $193_{28}$ | $98.43_{0.55}$ | $354_{29}$ |
| Our$_{Kling}$ | $\mathbf{1064_{125}}$ | $\mathbf{194_{24}}$ | $\mathbf{97.53_{2.41}}$ | $\mathbf{404_{40}}$ | $\mathbf{641_{103}}$ | $\mathbf{135_{27}}$ | $\mathbf{98.93_{0.41}}$ | $\mathbf{325_{30}}$ |
| | **Ablation Study** | | | | | | | |
| LaSR$_{Kling}$ | $1225_{137}$ | $211_{32}$ | $96.98_{2.10}$ | $451_{43}$ | $710_{113}$ | $151_{31}$ | $98.24_{0.53}$ | $341_{31}$ |
| Manual$_{Kling}$ | $1329_{129}$ | $218_{28}$ | $97.17_{2.33}$ | $477_{36}$ | $667_{127}$ | $164_{25}$ | $98.77_{0.38}$ | $357_{27}$ |
| GT$_{Kling}$ | $1022_{132}$ | $186_{31}$ | $97.88_{2.18}$ | $397_{34}$ | $633_{105}$ | $132_{24}$ | $98.36_{0.47}$ | $320_{34}$ |
| Simulator | N/A | N/A | N/A | N/A | $67_{12}$ | $16_3$ | $99.52_{0.01}$ | $48_{14}$ |

Table 2: Quantitative comparison of video forecasting. All reported metrics are averaged across physical systems. DA refers to DragAnything, MCtrl to MotionCtrl.

**Evaluation of Generated Videos.** We evaluate generated videos along two axes: visual quality and physics alignment. For visual quality, we use the Fréchet Video Distance (*FVD*) and Fréchet inception distance (*FID*) Unterthiner et al. (2019); Heusel et al. (2017) to measure the difference between generated and ground-truth videos. For physics alignment, we use AMT Li et al. (2023) to quantify *motion smoothness*, and use *TrajEr* Zhang et al. (2024) to measure the deviation between the input trajectory and the actual trajectory in the generated video.

We conduct pairwise human comparisons across models for both visual quality and physics alignment. For each model pair, three graduate-level annotators independently judged the better video. All comparisons are anonymized and randomized. Evaluations were conducted on ten videos per system across all physical systems. Annotators are provided with system descriptions to aid in assessing physics correctness. Human evaluation is restricted to top-performing models based on automatic metrics: CogVideoX1.5 (best trajectory-free baseline), Tora (best open-source trajectory-guided model), Kling (overall best trajectory-guided model), and physics simulator.

**Video Generation Models.** We conduct experiments on trajectory-free and trajectory-guided I2V models. For trajectory-free baselines, we consider state-of-the-art I2V models, including SVD Blattmann et al. (2023), CogVideoX1.5 Yang et al. (2025), Cosmos Team (2025), and Hunyuan-Video Kong et al. (2025). ID Chen et al. (2022) is an encoder-decoder model that generates videos frame-by-frame without trajectory guidance. For trajectory-guided I2V models, we use DragAny-

thing Wu et al. (2025), MotionCtrl Wang et al. (2024b), SG Namekata et al. (2025), Tora Zhang et al. (2024) and Kling (a commercial model) Team. These models are conditioned on the initial image and the given trajectory. For models that accept a single trajectory, we use the one with the highest motion magnitude, while for models that support multiple trajectories, we use the top-5 with the highest motion magnitudes. For synthetic data, we also use a physics simulator Huang et al. (2024) to generate future videos serving as a reference for upper-bound performance.

**Implementation Details.** We resize initial images to match each model's input resolution. The video length is fixed at 5 seconds, with frames per second (FPS) set per model requirements. For models requiring text prompts, we use either official prompt guidelines or generate prompts using GPT-4o Yang et al. (2025). Trajectories are normalized and scaled to match each model's spatial resolution and sampled uniformly at 2 points per second. All models are run on a machine with an NVIDIA 80G A100 GPU or API without fine-tuning.

| Pairs | Visual | Physical |
|---|---|---|
| Real | | |
| **Our**$_{Tora}$ vs. CogVideoX | 61% | 84% |
| **Our**$_{Kling}$ vs. Our$_{Tora}$ | 67% | 71% |
| **Our**$_{Kling}$ vs. Man$_{Kling}$ | 57% | 63% |
| Synthetic | | |
| **Our**$_{Tora}$ vs. CogVideoX | 78% | 77% |
| **Our**$_{Kling}$ vs. Our$_{Tora}$ | 66% | 61% |
| **Our**$_{Kling}$ vs. Man$_{Kling}$ | 59% | 64% |
| **Simulator** vs. Our$_{Kling}$ | 94% | 97% |

Table 3: Human evaluation via pairwise comparisons. In each row, the **bolded** model indicates the winner, and the following cell reports its win rate for each criterion.

**Results and Analysis.** Table 2 presents automatic evaluation results on all models. Models guided by trajectory consistently outperform trajectory-free baselines in both visual quality and physics alignment. Among trajectory-guided I2V models, Kling guided by trajectories predicted by our method achieves the best performance, closely approaching Kling with ground-truth trajectories. Real initial frame settings have lower performance than synthetic settings, likely due to background noises and systems complexity. In the synthetic setting, all models perform significantly worse than the simulator, indicating that current data-driven video generation models struggle to capture physical dynamics, even when guided by ground-truth trajectory. This highlights the need for future work to improve the physical alignment of data-driven approaches.

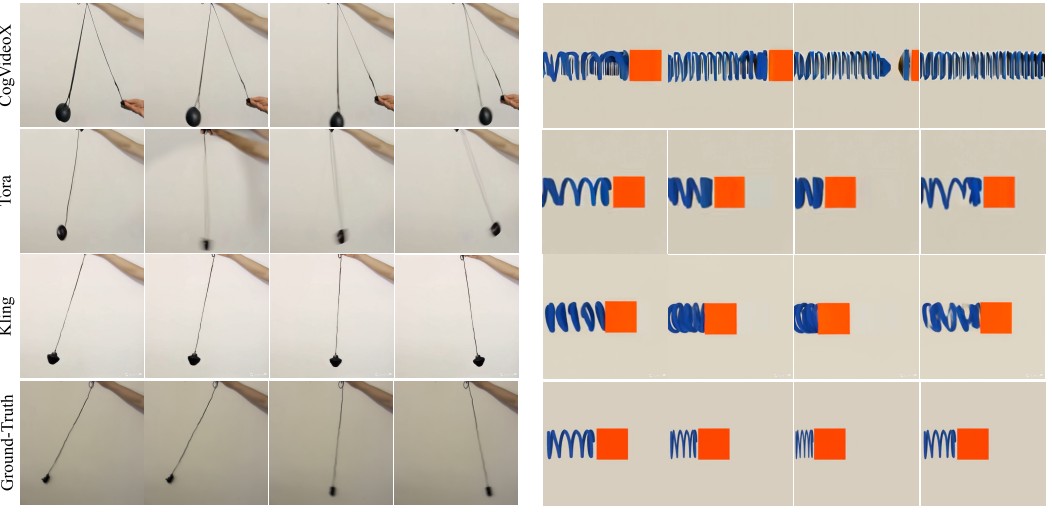

(a) Single pendulum system.    (b) Damped spring-mass system.

Figure 3: Qualitative comparisons across models.

For human evaluation in Table 3, annotators consistently preferred trajectory-guided models over trajectory-free baseline on both physical alignment and visual quality. Inter-annotator agreement was measured using Fleiss' Kappa Fleiss (1971), yielding a score of 0.73, which indicates substantial agreement among annotators. Notably, Kling with ReSR-guided trajectories was preferred over its manually guided counterpart, confirming that learned equations offer more accurate and reliable motion control. In the synthetic setting, the physics simulator was consistently rated as the most

physically accurate, highlighting the limitations of data-driven approaches. We attribute this to a fundamental difference: current video generation models are trained to capture statistical correlations in large-scale datasets, but lack explicit modeling of physical causality. In contrast, physics simulators generate motion directly from governing equations, ensuring high physical fidelity. However, simulators have their own limitations. They are not scalable across diverse scenarios and tend to lack realism when applied to real-world scenarios. This highlights the value of our method, which seeks to combine the interpretability and physical grounding of governing equations with the flexibility and realism of data-driven generative models.

Figure 3 illustrates qualitative comparisons. Trajectory-guided models exhibit improved global motion consistency, while trajectory-free models (e.g., CogVideoX) often produce erratic or implausible dynamics. Even the strongest model, Kling, fails to capture fine-grained physical details such as spring deformation, suggesting that while trajectory conditioning improves high-level motion, current models still lack the physical inductive biases needed for fine-grained dynamic synthesis.

## 5 RELATED WORK

**Equation Discovery from Video.** Chari et al. (2019); Luan et al. (2021); Tetriyani et al. (2024); Garcia et al. (2024) aim to extract physical laws of dynamic systems directly from video, using symbolic regression or ODE-based methods. However, many of these approaches impose strong constraints on the equation structure, such as assuming linearity, or focus solely on estimating parameters of pre-defined models. Huang et al. (2024) uses autoencoders to encode video sequences into low-dimensional latent vectors and attempt to learn system dynamics in that space. These latent variables often lack physical interpretability, and the resulting dynamics are not expressed as symbolic equations. In contrast, our approach employs symbolic regression to directly learn explicit symbolic equations, capturing physically meaningful variables that map time to object positions, thus ensuring interpretability and physical alignment.

**Physics-aligned Video Generation.** Millington (2007); Todorov et al. (2012); Bonnet et al. (2022); Kohl et al. (2024); Ohana et al. (2024); Lv et al. (2024) use physics simulators to ensure physical realism in video generation, where dynamics are modeled via hard-coded rules and equations. While highly accurate, these simulators are typically limited to specific domains, require hand-crafted scenario design. On the other hand, Blattmann et al. (2023); Wang et al. (2024a); Yang et al. (2025); Team (2025); Kong et al. (2025) use diffusion or autoregressive architectures to synthesize diverse scenes from image or text prompts but often lack physical consistency, leading to unrealistic object motion Motamed et al. (2025).

Trajectory-guided video generation is a motion-aware video synthesis framework where object movement is explicitly controlled by numerical trajectories, which are typically represented as sequences of $(x, y)$ coordinates over time Xing et al. (2025); Ho et al. (2020); Song et al. (2020); Wang et al. (2024b); Wu et al. (2025); Namekata et al. (2025); FU et al. (2025); Zhang et al. (2024). In prior work, trajectories are manually drawn, which does not ensure physical alignment. In contrast, we use learned equations from observational data to generate future trajectories, ensuring that the future object motion follows discovered physical dynamics.

## 6 CONCLUSION

We introduce a novel physics-grounded, inference-only framework for motion forecasting in trajectory-guided video forecasting, which employs ReSR for equation discovery. Experimental results demonstrate that our approach can reliably generate future motion trajectories closely matching equations derived from classical mechanics. Experimental results also highlight the limitations of current SOTA I2V models. Even with accurate trajectories, generated videos may deviate in fine-grained details such as velocity or deformation. Addressing these limitations requires advances in controllable video generation models. Overall, our work illustrates the potential of integrating interpretable equation discovery with I2V models and paves the way for applications in scientific discovery and simulations for robotics. An exciting next step is to apply our approach to multi-body systems involving collisions and contact dynamics. We expect this will require integrating symbolic regression with hybrid modeling frameworks to capture discontinuous transitions.

## ETHICS STATEMENT

As part of our study, we hired human evaluators to assess the visual and physical quality of generated videos. All participants were recruited voluntarily through an online platform and provided informed consent before participating. We ensured anonymity of all responses and did not collect personally identifiable information. Participants were compensated above the local minimum wage relative to task duration.

## REPRODUCIBILITY STATEMENT

We release code and data at `https://anonymous.4open.science/r/ReSR-0083/`. Experimental details can be found in section 4.1 and 4.2. All video generation models used in our work are either publicly available or accessible through APIs (*e.g.,* Kling).

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

## A    THE USE OF LARGE LANGUAGE MODELS

In this research, large language models are employed as tools to support writing. Specifically, I used LLMs to check and refine the grammar of drafts. In addition, LLMs were applied as debugging aids during code development. Importantly, all core research ideas, experimental designs, analyses, and conclusions presented in this thesis remain our own.

## B    QUALITATIVE EVALUATION

Figure 4 and 4 illustrate qualitative comparison on different physical systems across models.

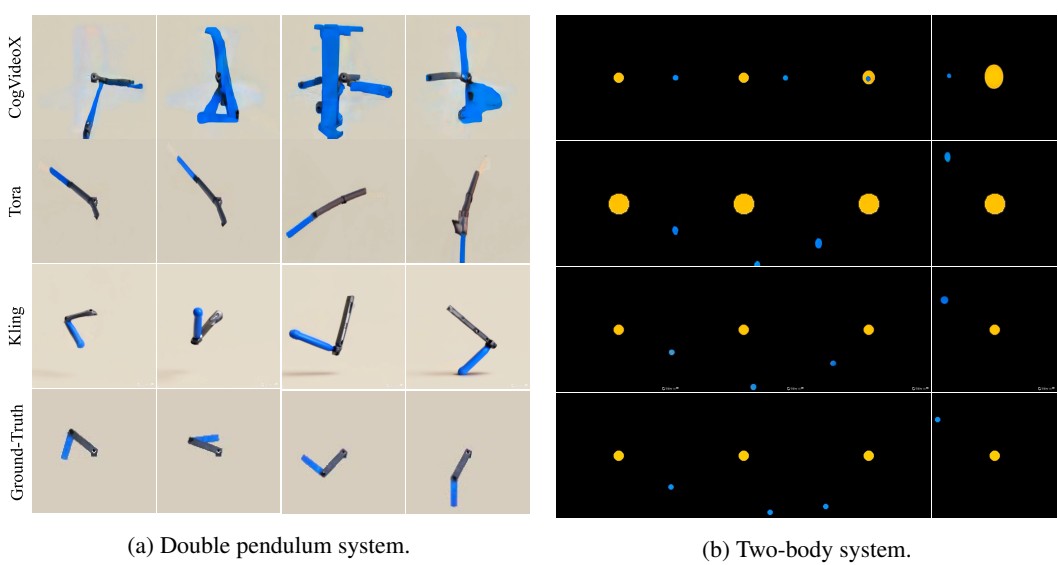

(a) Double pendulum system.    (b) Two-body system.

Figure 4: Qualitative comparisons across models.

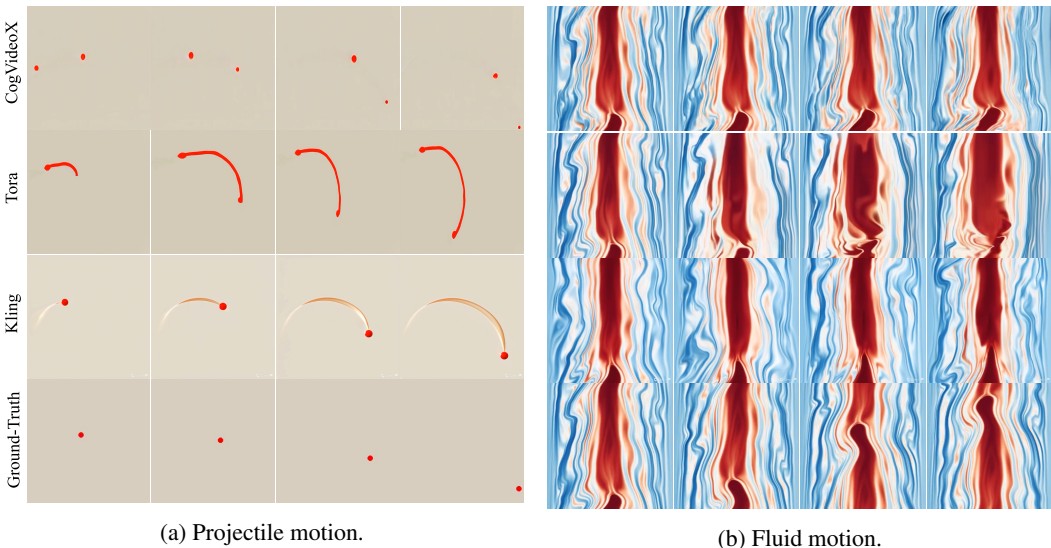

(a) Projectile motion.    (b) Fluid motion.

Figure 5: Qualitative comparisons across models.

