# OpenReview forum: "Physics-Grounded Motion Forecasting via Equation Discovery for Trajectory-Guided Image-to-Video Generation"
_ICLR.cc/2026/Conference — ICLR 2026 Conference Withdrawn Submission_

### Official Review · Reviewer_uc2J · 2025-10-27

**Soundness:** 2
**Presentation:** 2
**Contribution:** 2
**Rating:** 4
**Confidence:** 2

**Summary:**

The paper addresses the lack of physical realism in modern video generators by discovering explicit equations of motion from a short input video and then using the predicted future trajectory to control a trajectory-guided I2V model. The pipeline is: (i) extract object tracks from a short clip (CoTracker) → (ii) perform symbolic regression to recover a closed-form motion law, using a retrieval-based pre-training (ReSR) that initializes the SR population from a curated equation bank of physics laws → (iii) extrapolate the future trajectory with the discovered equation and drive an I2V model to synthesize future frames without generator fine-tuning. Experiments on classical mechanics scenes (spring–mass, pendulum, projectile) show that the method recovers ground-truth analytical equations and improves physical alignment of generated videos over baselines.

**Strengths:**

1. Clear pipeline; inference-only control ensures the improvements stem from the motion law rather than re-training the generator. The paper reports successful recovery of ground-truth equations and improved global motion consistency.
2. The overview and per-module descriptions are easy to follow; classical-mechanics scope and assumptions are explicit.

**Weaknesses:**

1. Experiments are limited to controlled lab videos of simple systems (single-object classical mechanics). It’s unclear how the method handles multi-object scenes, contacts/collisions, or non-rigid effects beyond the chosen examples.
2. Tracking noise/occlusions can corrupt trajectories; sensitivity of SR to noisy, short observations is not analyzed. (Method description assumes clean tracks.)
3. The attached showcase videos are not convincing to me. The quality of the video is low and some objects are distorted, shape changed.

**Questions:**

1. Can authors provide more convincing demos? Demos now can not support ReSR's contribution.
2. Please see the weaknesses above.

---

### Official Review · Reviewer_nLVP · 2025-10-30

**Soundness:** 3
**Presentation:** 1
**Contribution:** 2
**Rating:** 2
**Confidence:** 4

**Summary:**

Existing video generation models are based on statistical fitting and lack accurate physical alignment. This work introduces a novel retrieval-based pre-training method for symbolic regression, named ReSR, which motion-consistent equations from input videos. The authors then derive future trajectories from the recovered equations. These equations are subsequently used to forward with trajectory-guided image-to-video (I2V) models, enabling physically aligned video forecasting without the need to fine-tune the I2V models. Experimental results demonstrate that in classical mechanics tasks, such as spring-mass systems, pendulum swings, and projectile motion, ReSR can recover the analytical equations and generate videos with superior physical alignment compared to baseline methods.

**Strengths:**

- A better symbolic regression method.

The authors analyze the drawbacks of previous symbolic regression methods, such as PySR and LaSR, including large search spaces and slow convergence, and propose ReSR. This method is based on a carefully designed, pre-defined equation bank rather than random candidate equations, which improves convergence speed and achieves superior results, significantly outperforming the baselines.

- It demonstrates the performance of trajectory-guided video generation with physical alignment.

With the more physics-consistent trajectories generated by ReSR, trajectory-guided video generation can produce videos that better match real-world dynamics, providing a new approach for physically aligned video generation.

**Weaknesses:**

- The ability to generate physically-aligned videos without ground truth should be proven.

In this work, input videos are necessary to obtain analytical equations and predict future changes. This method cannot generate motion-consistent trajectories based solely on the initial frame, which limits its application scope.

- The method's extensibility has yet to be demonstrated.

In the introduction, the authors mention that "insights into object motion in classical mechanics can be easily extended into other types of motion." However, the experiments presented in the paper are still limited to simple laboratory settings and lack examples of applications in more general scenarios.

- bad writing.

In this paper, the authors exhibit several instances of improper writing, including but not limited to:

1、Inconsistent use of tenses.  In Sec.. 4.2, the phrase "Evaluations were conducted on ten videos per system across all physical systems." should use "are" instead of "were." The sentence "Inter-annotator agreement was measured..." mixes present and past tenses. There are other similar problems in this paper.

2、Incorrect singular/plural usage. In the introduction, "a handful of video clip" should be corrected to "clips."

3、Incorrect caption format. For the table type, the caption should be placed above the table, but in this paper, the caption is placed below the table.

**Questions:**

- Can the ablation study be expanded?

The paper compares the equation prediction performance of several methods, including PySR, KAN, and LaSR, but in Section 4.2, the authors only compare LaSR with Kling as the base model. Additionally, the authors do not explain how $Manual_{Kling}$ and $GT_{Kling}$ are generated, even in the supplementary materials.

---

### Official Review · Reviewer_Jxx2 · 2025-10-31

**Soundness:** 2
**Presentation:** 2
**Contribution:** 1
**Rating:** 2
**Confidence:** 3

**Summary:**

This paper proposes a neuro-symbolic framework that integrates symbolic regression with trajectory-guided image-to-video models to achieve physics-grounded video generation. The method extracts motion trajectories from input videos using CoTracker, discovers governing equations through a novel retrieval-based symbolic regression approach (ReSR) initialized with a curated physics equation bank, and uses the discovered equations to forecast future trajectories that guide video generation models without fine-tuning. The framework is evaluated on classical mechanics systems including spring-mass oscillators, pendulums, and projectile motion, demonstrating improved equation recovery and physical alignment compared to baseline methods.

**Strengths:**

1. Novel neuro-symbolic integration for video generation. The paper presents an innovative approach combining interpretable symbolic equation discovery with data-driven video generation models, bridging two typically separate domains. The ReSR method with retrieval-based pre-training from a physics equation bank is a well-motivated contribution that significantly improves convergence speed (44.3 iterations vs 61.4 for PySR) and equation accuracy (TED 0.80 vs 0.47) as shown in Table 1.
2. Comprehensive evaluation framework with meaningful metrics. The paper employs multiple complementary evaluation metrics including symbolic similarity (TED), trajectory error (MSE), convergence speed (ITB), and video quality measures (FVD, FID, smoothness, trajectory error), alongside human evaluation studies with substantial inter-annotator agreement (Fleiss' Kappa 0.73). The ablation studies systematically analyze the impact of initialization weight w and training/test split proportions.

**Weaknesses:**

1. Equation bank construction relies heavily on manual curation. Section 3.3 describes constructing the equation bank by manually adapting 106 Feynman equations through time-variable substitution where "time-dependent variables (e.g., velocity, acceleration, momentum) with the time variable t" and "variables that are independent of time (e.g., mass, density) are replaced with constant values (e.g., 10)."

2. The trajectory extraction pipeline uses heuristic selection. The paper extracts trajectories by sampling a uniform 10×10 grid and selecting "top-k trajectories with the highest motion magnitude" based on "temporal variance" (Section 3.2), motivated by the observation that "target objects in physics-driven videos typically exhibit the most motion." However, this heuristic may fail when background elements move more than objects of interest (e.g., camera motion, wind effects on curtains), or when the physical system involves multiple interacting objects with different motion scales.

3. Limited scope to idealized classical mechanics scenarios restricts practical applicability. The paper acknowledges in Section 4.1 that evaluation is limited to "classical physics systems" in "controlled laboratory environment" because it enables "direct evaluation against ground-truth equations." While this is reasonable for proof-of-concept, the scenarios tested (spring-mass, pendulums, projectile motion) represent highly simplified physics with analytical solutions.

**Questions:**

1. Your equation bank construction (Section 3.3) involves manually replacing time-dependent variables with "t" and time-independent variables with constants like "10". Can you provide evidence that this substitution process preserves the mathematical structure relevant for trajectory fitting?


2. Your trajectory extraction selects the "top-5 trajectories with the highest temporal variance" from a 10×10 grid (Section 3.2), but Table 2 shows substantially degraded performance on real initial frames versus synthetic ones (e.g., Kling: FVD 1064 vs 641, TraEr 404 vs 325). Can you quantify how often your heuristic correctly identifies the object of interest in your real-world test cases?

---

### Official Review · Reviewer_sRn7 · 2025-11-01

**Soundness:** 2
**Presentation:** 2
**Contribution:** 2
**Rating:** 4
**Confidence:** 3

**Summary:**

This paper introduces a neuro symbolic framework for generating physics grounded videos. The method operates at inferenc time by first extracting an object 2D motion trajectory from a short input video using a tracking model (CoTracker3). It then feeds this trajectory into a novel symbolic regression method, ReSR. It  enhances standard SR by initializing its search using candidate equations retrieved from a prebuilt equation bank of known physical laws. The retrieval is based on trajectory shape similarity using normalized dynamic time warping. The discovered symbolic equation is then used to forecast a physically consistent future trajectory. Finally, this forecasted trajectory is passed to an existing trajectory guided I2V model to synthesize the output video. The framework is evaluated on videos of classical mechanics systems, where it reports accurate equation recovery and improved physical alignment in the generated videos compared to baselines.

**Strengths:**

1. The core concept of connecting video analysis, symbolic equation discovery, and generative video models into a single pipeline is a novel approach to address the lack of physical realism in video generation.
2. An intermediant representation of the system is an explicit symbolic equation. This more explainable than black-box neural network predictors and provides genuine insight into the system's dynamics.

**Weaknesses:**

1. the method is exclusively evaluated on simple, 2D classical mechanics problems (springs, pendulums) in controlled, static-background lab settings. There is no evidence it can scale to complex, real-world 3D scenarios involving camera motion, occlusions, or non-trivial dynamics.
2. the ReSR method's success is predicated on the assumption that an equation similar to the true governing law already exists in its curated bank. This approach will not generalize to discovering novel physics or complex dynamics not represented in the bank.
3. the method operates entirely in 2D space . It does not model 3D motion, camera projection, or perspective, which are fundamental to any real-world video application.
4. the method of isolating the object of interest by selecting trajectories with the highest motion magnitude will fail in any video with camera motion or multiple dynamic background elements.
5. the retrieval step requires comparing the input trajectory to every equation in the bank via N-DTW. While parallelizable, this comparison is computationally inefficient and will not scale to a truly large, general-purpose bank of physical equations.
6. The framework is not scalable and couldn't fit into the scaling law.

**Questions:**

1. I understand that this is a pioneering work that use symbolic regression to represent physical effects. But it is still worth investigating that if this works in a in-the-wild enviroment where more noises are there in the video. Otherwise, if this work only applies to simple toy/lab env, then, physical engine might be a better choice?

---

### Note · Authors · 2025-12-31

I have read and agree with the venue's withdrawal policy on behalf of myself and my co-authors.